# Recent Synthetic Approaches towards Small Molecule Reactivators of p53

**DOI:** 10.3390/biom10040635

**Published:** 2020-04-20

**Authors:** Jerson L. Silva, Carolina G. S. Lima, Luciana P. Rangel, Giulia D. S. Ferretti, Fernanda P. Pauli, Ruan C. B. Ribeiro, Thais de B. da Silva, Fernando C. da Silva, Vitor F. Ferreira

**Affiliations:** 1Programa de Biologia Estrutural, Instituto de Bioquímica Médica Leopoldo de Meis, Instituto Nacional de Ciência e Tecnologia de Biologia Estrutural e Bioimagem, Universidade Federal do Rio de Janeiro, 21941-902 Rio de Janeiro, Brazil; giuliadiniz@hotmail.com; 2Departamento de Química Orgânica, Instituto de Química, Universidade Federal Fluminense, 24020-141 Rio de Janeiro, Brazil; carolgslima@gmail.com (C.G.S.L.); fernanda_pauli@hotmail.com (F.P.P.); ruancarlos@id.uff.br (R.C.B.R.); thaisbrito@id.uff.br (T.d.B.d.S.); fcsilva@id.uff.br (F.C.d.S.); 3Departamento Faculdade de Farmácia, Universidade Federal do Rio de Janeiro, 21941-170 Rio de Janeiro, Brazil; lprangel@pharma.ufrj.br; 4Departamento de Tecnologia Farmacêutica, Faculdade de Farmácia, Universidade Federal Fluminense, 24241-000 Rio de Janeiro, Brazil

**Keywords:** tumor suppressor p53 protein, wild-type p53, mutant p53, MDM2

## Abstract

The tumor suppressor protein p53 is often called “the genome guardian” and controls the cell cycle and the integrity of DNA, as well as other important cellular functions. Its main function is to trigger the process of apoptosis in tumor cells, and approximately 50% of all cancers are related to the inactivation of the p53 protein through mutations in the *TP53* gene. Due to the association of mutant p53 with cancer therapy resistance, different forms of restoration of p53 have been subject of intense research in recent years. In this sense, this review focus on the main currently adopted approaches for activation and reactivation of p53 tumor suppressor function, focusing on the synthetic approaches that are involved in the development and preparation of such small molecules.

## 1. Introduction

Cancer is a multifactorial disease characterized by the dysregulated growth of cells; tumor cells replicate and spread in a process called metastasis, which is the cause of most of the deaths related to such disease. In this context, the presence of tumor suppressor genes, which act as regulators of cell proliferation, are essential in the control of cancer progression. One of these genes is *TP53*, which is activated when DNA damage takes place, promoting the transcription of the p53 protein (wild-type—WTp53), which is responsible for arresting the cell cycle, among other functions [1,2,3].

Since its discovery in 1979, the p53 tumor suppressor protein has been the subject of intense research and is considered an important and challenging target in cancer therapy. The increase in the level of wild-type p53 in the cell nucleus promotes cell repair, senescence and/or apoptosis through different mechanisms [4,5,6,7,8]. However, in approximately 50% of all cancers, there is an inactivation of the p53 function through somatic mutations in TP53 [9,10].

Due to the association of mutant p53 with cancer therapy resistance, different forms of reactivation of p53 have been subject of intense research in recent years. Some studies focus on the insertion of second-site suppressor mutations, which partially restore specific DNA binding and/or stabilize the folding structure of the protein. Others involve the use of synthetic peptides derived from the p53 C-terminus with the aim of restoring the specific DNA binding and transactivation function to mutant p53 and inducing p53-dependent apoptosis in tumor cells [9,10]. It is known that p53 controls the cell cycle and the integrity of DNA, among other important cellular functions, but in the cancer context, its main function is to trigger apoptosis in tumor cells [11]. Similar to the strategy of rescuing the wild-type function of mutant p53, the idea of trapping on-pathway oligomers from the p53 aggregation reaction seems reasonable to avoid the building of the complex assemblies observed in cancer cells [11]. A drawback still related to this approach is the sparse information regarding p53 oligomers as the building blocks of these cancer assemblages. Answering whether heterogeneous oligomers are on- or off-pathway intermediates and investigating those involved in cancer through different methods, such as cryo-EM and super resolution microscopy, will definitely point oncogenic mutants of p53 as potential targets for anticancer therapy [11].

Another promising strategy for the restoration of wild-type p53 function in tumors that have lost p53 tumor suppressor activity is the identification of natural or synthetic small organic molecules that can reactivate mutant p53 to its wild-type version. As a result, a large number of research groups have been focusing in the last few years on the synthesis of small molecules with p53-related activity and important advances have been accomplished thus far [12,13,14,15,16,17,18].

As mentioned before, TP53 is mutated in around 50% of human cancers. Other cases include different reasons for p53 activity impairment, such as the oncogene MDM2 overexpression, which leads to WTp53 inactivation. The MDM2 (Murine Double Minute 2 or HDM2) and MDMX (Murine Double Minute X, HDMX or MDM4) are important regulators of p53, but their overexpression may lead to the inactivation of p53. In this sense, numerous tumor suppressor approaches are related to p53-MDM2/MDMX, namely preventing the formation of p53-MDM2 complexes, preventing the p53 protein ubiquitination degradation and modifying p53 transcriptional active region to stabilize the p53 protein [19,20,21,22]. It is important to mention that comprehensive literature reviews on the rationale of using small molecules and peptide that function as MDMX inhibitors or as dual MDM2/MDMX inhibitors have been recently published [23,24].

Other approaches for the reactivation of p53 have also been successful, for instance the inhibition of the nuclear export factor CRM-1 [25]. Another viable approach for the activation of p53 is the inhibition of casein kinase 1A1 (CKIa) [26,27], as well as of the sirtuins SirT1 and SirT2, which are implicated in a series of essential cellular processes and disease conditions [28]. Additionally, considering that a significant fraction of mutant p53 is found in a self-aggregated amyloid-like state in cancer cells [29,30,31,32,33], and these destabilized oncogenic mutants in the aggregated state are formidable targets for stabilization by drugs [11,31,32,33]. With this aim in mind, several strategies were designed in order to recapitulate p53 function in cancer. Overall, they include the use of small molecule or peptide stabilizers of misfolded p53, zinc administration, gene therapy, metallochaperones, alkylating and DNA intercalators, [34] blockage of p53-MDM2 interaction, impaired reactive oxygen species (ROS) detoxification and other p53 regulators [17,18,33,35].

In this sense, this review will focus on the recent currently adopted methods for the activation and reactivation of the p53 tumor suppressor function with an emphasis on the synthetic approaches to obtain the small molecules used as reactivators. Although most of the mechanisms involved in such processes are still not completely known, this review will approach the reactivation of p53 via two main pathways: (1) The reactivation of mutant p53 via the covalent reaction with thiol in the cysteine residues and other approaches and (2) reactivation of wild-type p53 via the antagonists of the p53-MDM2/MDMX interactions. Lastly, additional approaches such as CRM1, CKIa, and SirT1/SirT2 inhibition will be discussed.

## 2. Targeting Mutant p53: Restoration of the p53 Function

### 2.1. Quinuclidinone Derivatives

It is already known that mutations in the central core domain of p53 can lead to the formation of a pocket in L1/S3, affecting specifically cysteine (Cys) residues Cys124, Cys135, Cys141, Cys182, and Cys277. In this sense, several small molecules can interact with this pocket and cause small conformational changes that can reactivate mutant p53, while others such as Michael acceptors may react via alkylation, causing permanent and significant changes to the structure of the protein, also restoring its DNA binding activity [36,37,38]. Although this alkylation approach can reportedly reactivate p53 function, the mechanism of this process is still not completely known. However, it has been reported that cysteines 124 and 277 are the main target for R175H mutant reactivation [39].

In view of these facts, the synthesis of low molecular weight molecules that can promote the alkylation of p53 and restore its activity has become particularly popular, especially since the 2000s. One of the pivotal studies in this field reported the discovery of PRIMA-1 (APR-017) through the screening of a database at the Developmental Therapeutics Program of the National Cancer Institute (NCI). Although other mechanisms may also take place, both PRIMA-1 and its methylated version (PRIMA-1^MET^, also known as APR-246) are both converted to methylene quinuclidinone (MQ), which is a Michael acceptor that reacts covalently with cysteine residues in p53 (WT or mutant), leading to the reactivation of mutant p53 into a WTp53-like conformation, to induce cell apoptosis [36,39,40,41,42]. Indeed, a phase Ib/II clinical trial is currently ongoing to evaluate the side effects and best dose of PRIMA-1 analog APR-246 when given along with azacitidine in treating patients with mutated TP53 mutant myeloid cancers, as well as a phase III study to compare the rate of complete response (CR) and duration of CR in patients with TP53-mutated Myelodysplastic Syndromes [43]. Recently, our group demonstrated that PRIMA-1 acts by inhibiting mutant p53 aggregation [44]. We were able to elucidate the molecular mechanism through which PRIMA-1 rescues amyloid aggregates of mutant p53 and thereby reduces the dominant negative (DN) and gain-of-function (GoF) effects, indicating that mutant p53 aggregation is an excellent target for the development of new anti-tumoral drugs [44].

The synthesis of PRIMA-1 is reported in patents, in which the quinuclidinone hydrochloride **1** is used as the starting material (Scheme 1). The synthesis of PRIMA-1 is achieved by the treatment of **1** with an excess of formaldehyde and potassium carbonate. As for PRIMA-1^MET^, it is also synthesized from **1**, but in this case, **1** is treated with 1 equivalent of formaldehyde and potassium carbonate to form **2**. The dehydration of **2** gives compound **3**, which, upon treatment with sodium methoxide, affords **4** (not isolated), followed by a treatment with formaldehyde, which delivers PRIMA-1^MET^ [45].

Other types of α,β-unsaturated compounds such as maleimide derivatives can also be used to reactivate the ability of p53 to perform DNA binding and preserve its active conformation [46]. MIRA-1, for example, is a maleimide-type compound with low molecular weight that has been identified in screenings at the NCI data bank as a potential candidate for reactivating mutant p53-dependent cells in different human tumor cells, being more potent than PRIMA-1 in the induction of cell death [47]. Additionally, other structural analogues of MIRA-1 were tested in the inhibition of p53-dependent growths and showed similar activity (MIRA-2 and MIRA-3, Scheme 2). Interestingly, when the saturated analogues were evaluated, those did not present significant results, which clearly demonstrated that the unsaturated maleimide system is of utmost importance for the activity of such compounds [47]. Although very effective for p53 reactivation, MIRA-1 was shown to be highly toxic to different types of normal cells and demonstrated an acute toxicity which does not involve p53, inducing apoptosis in normal cells through a caspase-9-dependent pathway [48], which demonstrates the need for new derivatives aiming to reduce these toxic effects.

### 2.2. Maleimide Derivatives

The synthesis of MIRA-type compounds has been known since the 1960s. It is based on maleimide as the building block, which is commercially available or may be easily obtained from the oxidation of pyrrole or by the acid-catalyzed cyclization of maleimide [49]. In this context, Tawney and co-workers explored the synthesis of several maleimide derivatives using fairly simple reactions and producing a huge diversity of compounds. Overall, the amine group of maleimide is reactive under basic conditions, and may undergo, for instance, methylation reactions with formaldehyde, producing the intermediate MIRA-2; this intermediate can easily react with phosphorus trichloride to produce *N*-chloromethylmaleimide **5** (80% yield), as shown in Scheme 2b. The reaction of **5** with anhydrides forms acetoxylmethylmaleimides (e.g., MIRA-1 and MIRA-3) [50,51].

### 2.3. α,β-Unsaturated Carbonyl Compounds

The positive results obtained with the compounds of the PRIMA and MIRA series stimulated the search for other small molecules with the ability to reactivate mutant p53 through covalent binding to its central domain to restore its tumor suppressor function. For instance, the open-chain version of the maleimide ring in the MIRA derivatives leads to α,β-unsaturated compounds capable of acting as Michael acceptors. With this perspective in mind, derivatives containing the 3-benzoylacrylic acid moiety have been synthesized and used as Michael acceptors. The 3-benzoylacrylic acid **6** and its fluorinated (*E*)-4-(4-fluorophenyl)-4-oxobut-2-enoic derivative **7** (Figure 1) raised the melting temperature of the core domain of WT p53 and the hotspot mutants R175H, Y220C, G245S, R249S, and R282 by 3 °C, which implicates on a thermodynamic stabilization of the mutant into a WTp53-like conformation [34]. Interestingly, in this work the authors were able to determine the order of reactivity of the different cysteine moieties in mutant p53 structure.

Another class of compound that has shown promising results are *N*-alkylated and *N,N*-dialkylated 3,5-bis-(arylidene)piperidones, which are obtained from 4-piperidone [52]. Such molecules have been shown to have high activity against the A549 resistant human lung carcinoma cell line and are structurally related with the natural compound curcumin, which is the main component of the rhizome of *Curcuma longa* and has proved to be a powerful chemopreventive and anticancer agent, with several other reported biological activities. Further research in this area identified a series of synthetic curcumin analogs containing two arylidene-α-carbonyl units and a nitroxide group [53]. These compounds, which bear two structural antioxidant moieties, showed efficacy and selectivity in killing cancer cells A2780, MCF-7, and H9c2 by converting mutant p53 to a transcriptionally active WTp53-like form [54]. The central strategy was to combine the anticancer properties of curcumin derivatives and the antioxidant capacity of the nitroxide groups, which has the ability to reduce damage caused by ROS. Among the synthesized compounds, a very active one was identified and denominated as HO-3867. The synthetic route to HO-3867 involves a Claisen–Schmidt condensation between 4-piperidone hydrochloride **9** and *p*-fluorobenzaldehyde **8** giving **10**, which leads to the product HO-3867 after alkylation with substituted pyrrol **11** (Scheme 3).

### 2.4. Pyrimidine, Quinazoline, and Quinoline Derivatives

Also considering that small molecules that act as Michael acceptors have the ability to react covalently with cysteine residues of mutant p53 without compromising its binding to DNA, Fersht et al. demonstrated that 5-chloro-2-(methylsulfonyl) pyrimidine-4-carboxylic acid, the so-called PK11000, increases the thermal stability of mutant p53 by reacting with cys182 or cys227 (Scheme 4a) [55,56]. Sulfonylpyrimidines are electrophilic compounds capable of reacting with nucleophiles, for example thiols, forming structures such as **13**; indeed, K11000 has shown to be a potent and selective alkylating agent, and is currently being targeted in a preclinical investigation for the therapy of patients with triple negative breast cancer [57]. PK11000 is a commercially available compound and its synthesis can be accomplished via the reaction of dichlorooxobutenoic acid **14** with thiocompound **15** followed by oxidation of intermediate **16** with hydrogen peroxide (Scheme 4b) [58].

In a similar path, styryl quinazoline compound CP-31398, which contains a double bond conjugated between a benzene and a quinazoline ring, has also been able to stabilize the active conformation of mutant p53 [59]. This compound inhibits the growth of small human tumor xenografts in vivo and in tobacco-induced lung cancer. Recently, it was demonstrated that CP-31398 is a p53-modulating agent with potential to act as a chemopreventive agent for different cancer types [60,61,62,63]. The synthesis of CP-31398 was reported by Sutherland and co-workers in 2012 (Scheme 5); in this route, the condensation reaction between quinazolinone **17** and 4-methoxybenzaldehyde **18** in the presence of sodium acetate gives the styryl quinazoline **19**, which after a chlorination/substitution reaction sequence leads to CP-31398 [64].

Another very promising compound for the reactivation of p53 via covalent modification is 2-vinylquinazolin-4-(3*H*)-one, the so-called STIMA-1 (Scheme 6) [65]. Similarly, the activity of this compound is related to the exogenous olefin group that can act as a Michael acceptor. Zach and co-workers found that STIMA-1 is more potent than CP-31398 in suppressing the growth of mutant p53-containing tumor cells [65]. In this sense, there are two reported synthetic approaches towards STIMA-1 using anthranilic acid derivative **20** as starting material. In the method developed by Witt and Bergman, 3-chloropropionyl chloride **21** is used to prepare 2-(3-chloropropionylamino)-benzamide **22**, which upon cyclization and dehydrohalogenation with sodium hydroxide in ethanol gives STIMA-1 (Scheme 6) [66]. Another approach was described recently by Abe and co-workers, and consists in a one-pot copper-catalyzed Ritter-type reaction with acrylonitrile **23** to achieve the same molecule (Scheme 6) [67].

P53R3 is a quinazoline-type molecule which was originally identified using p53 DNA binding assays [68]. This compound restores sequence-specific DNA binding to several p53 hot spot mutants, but also improves the recruitment of wild-type p53 to target gene promoters. The synthesis of P53R3 was described back in 2007 in a patent by Mallams and co-workers (Scheme 7) [69]. 2,3-Dihydroquinazolinone **26** is obtained through the cyclization reaction between propargyl chloride **24** and ethyl 2-aminobenzoate **25**. Next, **26** is reacted with piperazine to form intermediate **27**, which gives **29** after treatment with 4,4’-(chloromethylene)bis(chlorobenzene) **28** in basic medium. The chlorination of **29** then leads to **30**, which is converted to P53R3 after reaction with L-valine methyl ester **31**.

In a similar path, SCH529074 is another quinazoline-type molecule which can reactivate mutant p53 and also bind to p53 DNA binding domain, interrupting HDM2-mediated ubiquitination of WTp53 [15,70]. SCH529074 synthesis can also be achieved using intermediate **27**; the reaction of **27** with benzyl chloride gives **32**, which is next chlorinated, furnishing **33** (Scheme 8) [69]. **33** is then reacted with *N*,*N’*-dimethylpropane-1,3-diamine, affording **34**, which is next converted to **35** after a reduction step. Finally, the reaction of **35** with 4,4’-(chloromethylene)bis(chlorobenzene) **28** gives rise to the product.

COTI-2, a thiosemicarbazone derivative, is another molecule which has showed promising activity in reactivating p53 (Scheme 9) [71]. Indeed, a phase 1 clinical trial in gynecological cancers is currently ongoing [72]. Although the mechanism of action of COTI-2 is not yet completely known, it has been already shown that this compound binds both to full length and DNA-binding domain of mutant p53, leading to the refolding of the mutant protein [73]. COTI-2 can be synthetized from the reaction of 1,1’-thiocarbonyldiimidazole **36** with *N*-(2-pyridyl) piperazine **37**, giving **38** as intermediate. Next, **38** is reacted with hydrazine to furnish **39**, which undergoes an addition reaction with 6,7-dihydro-5*H*-quinolin-8-one, giving the desired product [74].

### 2.5. Pyrrol and Pyrazole and Derivatives

Recently, Fersht and co-workers identified a family of hybrid 1*H*-pyrrole-1*H*-pyrazole heterocycles from which compound PK7088 was highlighted, being biologically active in vitro against cancerous cells carrying the Y220C mutant p53 protein. Assays indicated that the PK7088 molecule increased the amount of WTp53 levels and restored its transcriptional functions and apoptosis [75]. This compound was identified in a library obtained by combinatorial chemistry, and was synthesized in four steps from 2-(4-fluorophenyl) acetonitrile; the key step was the synthesis of the pyrazole ring from 2-(4-fluorophenyl)-3-oxopropanenitrile via condensation with hydrazine. As for the construction of the pyrrole ring, it was achieved using the classic Paal-Knorr method from **40** (Scheme 10).

Prodigiosin is a tripyrrole red pigment member of the prodiginine family; it is a secondary metabolite of the human pathogen *Serratia marcescens* [76]. This compound has been receiving growing attention owing to its reported antimicrobial, immunosuppressive, and anticancer properties. In 2014, El-Deiry and co-workers reported that prodigiosin is a promising p53 reactivator, restoring a deficient p53 signaling pathway and producing antitumor effects via a dual mechanism which involves p73 upregulation and disruption of the mutant p53/p73 complex. The first report on the total synthesis of prodigiosin dates back to 1962 [77], but other approaches have also been reported [76]. Tripathy, Lavallée and co-workers, for instance, reported the synthesis of fragment **42** via the reaction of 4-methoxy-3-pyrolin-2-one **41** with DMF in the presence of POBr_3_ followed by the Suzuki cross-coupling between **42** and boronic acid **43** (Scheme 11) [78]. Finally, fragment **43** is coupled with pyrrol **45** in acidic medium, giving prodigiosin as product.

### 2.6. Zinc Metallochaperones

Zinc plays a crucial role in the structure and properties of p53, since this protein binds to DNA through a zinc-stabilized structurally complex domain [79]. Considering these concepts, D’Orazi’s group investigated thoroughly the purpose of zinc in p53 reactivation in mutant p53-expressing cancer cells, as reported in 2011 [80]. The group observed that zinc partly induced the transition of mutant p53 into a functional conformation, being able to re-establish chemosensitivity in breast cancer cell lines expressing the R175H mutation, as well as in glioblastoma ones expressing the R273H mutation.

This study paved the path for a series of works involving a new class of compounds, the zinc metallochaperones (ZMCs), which have appeared as promising candidates for restoring p53 [81]. This new type of anti-cancer drug acts by targeting a precise set of zinc-binding p53 mutations. Carpizo’s group has very recently described the use of such types of compounds to treat BRCA1 deficient breast cancer and found very interesting results (Scheme 12) [82]. The compound ZMC1, which is commercially available, combined with olaparib, was very effective in inhibiting tumor growth, while its complexation with zinc (Zn-1) showed improved efficacy.

Another interesting p53 reactivator is the case of a bifunctional ligand L^H^. The compound presents zinc metallochaperone features and strongly interacts with mutant p53. The simple insertion of an iodine atom to the compound structure (Figure 2) promotes inhibition of mutant p53 aggregation, restores zinc binding to mutant p53, and reactivates WTp53 transcriptional function. The effects were observed both in vitro and in tumoral cells. Also, the ligand presented minimal toxicity to non-cancerous organoids, showing a selective cytotoxicity to mutant p53 tumors [83].

### 2.7. Other Classes of Compounds for the Reactivation of Wild-Type p53

Inspired in natural products such as styryl lactones, which are known to present high cytotoxicity and are found in the *Annonaceae* plant gender, Kondaiah and Prasad’s group reported in 2013 the discovery of MPK-09, another promising compound that displays antitumor activity. Such molecule showed to be very selective and highly potent in the restoration of p53 functions of the mutants R175H, R249S, R273H, and R273C [84]. Although the mechanism through which MPK-09 reactivates p53 is not completely known, it is conceivable that its α,β-unsaturated dicarbonyl moiety may act as a Michael acceptor toward thiols (Scheme 13) [85].

MPK-09 can be synthetized via a 5-step high yielding route from 2,2-dimethyl-1,3-dioxolane-4,5-dicarboxamide **46**. Initially, **46** is converted into **47** after the reaction with a Grignard reagent, followed by a reduction step to afford **48**. Next, an alkylation reaction gives **49**, which is converted to **50** after the reaction with dimethyl methylphosphonate, and finally undergoes a Wittig-Horner olefination, finally giving MPK-09 as product.

PK5174 is also a noteworthy compound with good affinity for the Y220C mutant p53 that inhibits its aggregation by acting on specific points in the p53 core domain (Figure 3). Interestingly, in this compound the ethynyl group is a nonclassical bioisostere to iodine, forming halogen bonds within the protein pocket [86,87]. Chetomin (CTM), a natural product extracted from the fungus *Chaetomium cochliodes*, which has antibacterial and antifungal properties [88], restores the Hsp40-p53 axis in a R175H mutant model and p53 MDM2-dependent degradation (Figure 3) [89]. Thiosemicarbazones are also a very important class of compounds, with some of them having reached clinical trials stage. Compound NSC319726, for instance, has restored effectively the function of R175H mutant p53 with extensive apoptosis induction [90].

Additionally, through a virtual search based on a set of information on the pocket L1/S3 of the p53 protein structure, stictic acid (also known as NSC87511) was selected as a potential p53-reactivating compound by targeting cysteine 124 (Figure 3). Indeed, it has accomplished the reactivation of R175H mutant p53 of human osteosarcoma cells, being even more potent than PRIMA-1 [8]. Remarkably, its structure bears little resemblance to the conventional reactivators of p53 function. It is important to note that among all of the small molecules capable of restoring the function of p53, this natural product is the only one that does not bear a nitrogen atom. Resveratrol is another molecule that has shown to inhibit the aggregation of p53 mutants in vitro, in tumor cells and in xenotransplants implanted in a nude mouse model [91]. High doses of resveratrol (100 µM) were required to exert the anti-amyloid and antitumoral effects. There are great hopes that synthetic chemistry will lead into resveratrol analogs with higher potency that can be used as pharmaceuticals.

RETRA is another small-molecule which have shown to activate a series of p53- regulated genes, specifically suppressing mutant p53-bearing tumor cells in vitro and in mouse xenografts, as reported by reported by Chumakov’s group in 2008 (Figure 3) [92]. The treatment of mutant p53-expressing cancer cells with this compound has resulted in a considerable increase in the expression level of p73, as well as a discharge of p73 from the blocking complex with mutant p53. Importantly, RETRA has shown to be active against tumor cells expressing several p53 mutants, still not affecting normal cells. Later in 2015, Sonnemann and co-workers described the activity of RETRA against Ewing’s sarcoma cells and observed its activity was not dependent on their TP53 status, being effective against p53-deficient and p53 wild-type cell lines [93].

## 3. When p53 Is Inactivated by Oncogenes: Small Molecules Antagonists of p53-MDM2/MDMX

MDM2 (murine double minute 2), also known as E3 ubiquitin-protein ligase MDM2 or HDM2, is an important negative regulator of WTp53 in normal cells. In tumors, it acquires an oncogenic role due to its overexpression that leads to WTp53 inactivation. This biological target has been highly investigated to restore p53 activity, and also has the largest set of small molecule inhibitors. Moreover, MDM2 is the therapeutic target with the highest number of substances in clinical trials against various types of cancer in humans [94]. Thus, the disruption of the MDM2-p53 protein complex is a promising strategy for the treatment of various types of cancer via the restoration of WTp53 function. Indeed, a compilation of patents on small molecule MDM2 inhibitors has been recently published [24] and a critical discussion on the biological rationale of these strategies have also been conducted [23,95].

### 3.1. Cis-Imidazoline Derivatives

Back in 2004, 1,2,4,5-tetrasubstituted-4,5-*cis*-imidazolines, also known as Nutlins, were discovered via a high-throughput screening at Hoffman-La Roche. This family of compounds are potent and specific inhibitors of p53-MDM2, which have inspired the discovery of more selective MDM2 inhibitors [96,97,98]. In this family, Nutlin-2 (IC_50_ = 140 nM) and Nutlin-3a (IC_50_ = 90 nM) were the compounds with the highest affinity for the recombinant p53-MDM2 protein. Crystal structures of the complex of Nutlin-3 with p53-MDM2 revealed that the two 4-chlorophenyl rings interact with the residues Trp23 and Leu26 pockets and the isopropoxy group of the other benzene ring interacts in the Phe19 pocket (Figure 4). This type of interaction with three distinct pockets came to be known as the “three finger pharmacophore model”.

The synthesis of Nutlins was initially reported in patents by Hoffmann-La Roche, in which the authors describe their synthesis from benzil **51** as a racemic mixture (Scheme 14) [99].

Nutlin-3a can be obtained through the condensation of the substituted 1,2-diamine **52** with aromatic ester **53** in the presence of trimethylaluminum to give the imidazoline **54**. Next, **54** is treated with phosgene in the presence of a base such as triethylamine to give the racemic carbamoyl chloride **55**, which can be purified using chiral supercritical fluid chromatography (SFC). The coupling of the racemic carbamoyl chloride **55** with 2-oxopiperazine provides the Nutlin-3 as a racemic mixture; the enantiomers were then separated by SFC [100,101].

More recently, Johnston and co-workers reported the first enantioselective synthesis of (−)-Nutlin-3 [102]. The authors developed highly diastereo- and enantioselective aza-Henry additions, i.e., carbon–carbon bond-formation addition of aryl nitromethanes to imines, affording adducts that were used to prepare the *cis*-stilbene diamine key intermediate precursor (Scheme 15). The aza-Henry adduct **58** was obtained via the reaction of *N*-Boc-protected imine **56** with aryl nitroalkane **57** and chiral bisamidine catalyst BAM. Next, the reduction of the nitro group afforded amine **59**, which was acylated with 2-isopropoxy-4-methoxybenzoic acid giving **60**, and later submitted to standard TFA-promoted Boc removal for the obtainment of **61**. Then, the monoamide **62** was achieved by a carbamoylation reaction with 2-oxopiperazine; finally, the Nutlin-3a compound was obtained after a Hendrickson’s dehydrative cyclization [103,104].

Later on, the same group reported a further optimization in the synthetic route with the use of a novel monoamidine organocatalyst (MAM) that provided high enantioselectivity at a higher reaction temperature (−20 °C) employing low catalyst loadings. This additional refined approach led to the synthesis of (−)-Nutlin-3 in a 17 g batch scale with only three chromatographic purification steps [105].

More recently, Fantinatti and co-workers developed the synthesis of Nutlin-3 as a scalemic mixture in which the levorotatory isomer (−)-Nutlin-3 was obtained in 84:16 ratio (Scheme 16) [100]. Additionally, the synthesis of enantioenriched (+)-Nutlin-3 was discribed using catalyst **64**, which was obtained using (*1S*,*2S*)-cyclohexanediamine as starting material (Scheme 16). The synthetic pathway in this case was initiated with 4-chlorobenzaldehyde **63**, which leads to diamine **52** after the reaction with ammonium acetate; next, amide **61** is synthetized from **52** in the presence of **65** and catalyst **64**. After two additional reaction steps, (+)-Nutlin-3 is obtained as a scalemic mixture. It is also worth mentioning that an array of biological tests showed that the activity of scalemic (−)-Nutlin-3 was comparable to that of the commercial eutomer in activating the p53 pathway [104].

Following a similar path, many new compounds were synthesized based on the structure of Nutlin-3a, with some of them displaying promising results. Vu and co-workers, for instance, reported a new member of the Nutlin family of MDM2 inhibitors, the so-called RG7112 (Scheme 17) [106,107,108]. This compound has advanced to phase I clinical trials in combination with cytarabine in patients with acute myelogenous leukemia, neoplasms and hematologic neoplasms [104]. Basically, as a strategy to optimize the structure of the original Nutlin compounds, the authors chose to preserve the most important features for binding to MDM2, e.g., the 4-chlorophenyl and isopropoxy groups, while exploring alterations in other sites of the molecule. Three main changes in the Nutlin structure were proposed: (a) methyl groups were placed at the imidazoline ring to prevent its oxidation; (b) a variety of polar groups were inserted into the amide side chain to explore their possible effects on binding and pharmacokinetic properties; and (c) a replacement of the 4-methoxy group for *tert*-butyl, owing to in vitro metabolism studies, which identified this as the most labile site in the molecule. The preparation of such compounds is exemplified in Scheme 17, which shows the synthesis of RG7112 [109]. Initially, the imidazoline **68** is obtained by the reaction of diamine **66** and benzoate ester **67** in the presence of trimethylaluminum. Next, the intermediate **69** is obtained by the phosgenation of **68**. Then, the reaction of **69** with substituted piperazine **70** in the presence of triethylamine leads to racemic compound RG7112. The enantiomers are later separated by chiral chromatography.

Although compound RG7112 showed high affinity with the p53-MDM2 protein, the results of the phase I clinical trial of this compound alone or in combination with cytarabine in patients with liposarcoma showed limited reduction in tumor volume. Currently, a clinical trial with RG7112 combined with doxorubicin is ongoing [109].

In 2010, Domling’s group introduced a new process for predicting and preparing antagonists of the p53/HDM2 interaction [110]. This approach consisted on the use of multicomponent reactions (MCR) to synthesize new potential p53-HDM2 inhibitors starting from a fragment. Although no protein–protein interaction analyses were conducted, computational studies have indicated the potential of the new molecules, which imidazole PB11 being the most promising one. The synthesis of the compound PB11 was achieved starting with the Orru three-component reaction between *p*-chlorobenzaldehyde **63**, isocyanoacetate **71**, and cyclopropylmethylamine **72** followed by an amidation step (Scheme 18).

### 3.2. Spiro-Oxindole, Indole, and Carbazole Derivatives

Since the discovery of the imidazolines or Nutlins family, many research groups have invested in the search for new nitrogen heterocycles that include benzodiazepines [111], indole-2-carboxylic acid derivatives [97] and pyrrolidines [112], among others. Wang and co-workers, for example, reported the design of a spiroindoline-3,3’-pyrrolidine series as potent, highly selective and orally active small-molecule inhibitors of the MDM2-p53 interaction [98,113]. The authors identified that the natural alkaloids spirotryprostatin A and Alstonisine displayed a steric hindrance that would keep them from fitting properly into the MDM2 cleft, but presented features in their structure that could be used as inspiration for designing new MDM2 inhibitors (Scheme 19).

Considering that the p53-MDM2 crystal structure showed that Phe19, Trp23, and Leu26 hydrophobic residues in p53 interact with a hydrophobic cleft in MDM2, the authors designed small molecules that could block this interaction [112]. Specifically, the oxindole scaffold could mimic the Trp23 side chain of p53 by providing hydrogen-bonding and hydrophobic interactions with MDM2, while the spiropyrrolidone ring would provide a rigid system that could bear hydrophobic groups that could resemble the side chain of Phe10 and Leu26. In this sense, based on an early approach published in 2000 by Williams’ group for the synthesis of alkaloids [114], Wang’s group reported an approach having an asymmetric 1,3-dipolar reaction as the key step to synthetize spiro-oxindoles that was later used in the synthesis of a series of highly active MDM2 inhibitors [115]. In this sense, in 2005, Wang’s group reported the synthesis of a series of compounds (**78a**–**f**, Scheme 19), which had their activity evaluated in WTp53 LNCaP human prostate cancer cells. Remarkably, compound **78d** proved to be a potent MDM2 inhibitor, being efficient in inhibiting cell growth with an IC50 value of 0.83 µM and also presenting low toxicity for normal cells [98].

Although compound **78d** proved to be highly active and selective, it still had a low potency when compared to the most potent peptide-based inhibitors [98]. Considering this premise, Wang’s group conducted further structural optimizations and reached compound MI-63, which showed to be highly potent with a K_i_ of 3 nM, being 2000 times more active than the natural p53 peptide (Figure 5) [116]. Later on 2008, seeking to improve MI-63 PK profile, which was unsuitable for in vivo evaluation, the same group discovered a compound called MI-219 (Figure 5), which also contains a *tert*-butyl group in the spiropyrrolidone moiety but has a different substitution pattern in the amide group. MI-219 indeed has shown promising results, being highly active in p53 reactivation and presenting very low toxicity [117]. Another very interesting compound in this series was MI-147, which showed very low values of Ki, being a highly potent inhibitor; MI-147 was able to selectively activate p53 in SJSA-1 cell lines and to dose-dependently induce cell death, being also highly efficient in inhibiting tumor growth in the SJSA-1 xenograft model.

Further structural optimization of MI-219 led to MI-888 (Scheme 20), in which the 1,2-diol moiety of the molecule was constrained by the insertion of a cyclobutyl ring, which would block this metabolic oxidative site [118]. In addition, the fluorine atom was removed from the oxindole ring, leading to an enhanced MDM2 binding affinity and improved pharmacokinetic profile. MI-888 showed to be stable in a variety of solvents, but when treated in the presence of MeOH or MeCN/H_2_O, formed four products that were later identified as stereoisomers [119]. In that way, the isomerization of MI-888 was further verified and the different stereoisomers had their p53-restoring activity evaluated. Interestingly, the different isomers presented markedly different activity as MDM2 inhibitors. An isomerization mechanism was proposed in which the *trans-cis* isomer is initially formed and undergoes a ring-opening reaction in the presence of MeCN/H_2_O, giving an imine intermediate **79**. The intramolecular ring-closure reaction of imine **79** leads to the formation of the three remaining isomers (Scheme 20).

It is known that MI-219 has a clear mechanism-based antitumor activity both in vitro and in vivo, however, high doses were required, whereas MI-888 failed to achieve complete tumor regression. Therefore, Wang and co-workers strived to further optimize the structure of such compounds and discovered MI-77301, also known as SAR405838 (Figure 4) [120]. Still presenting a structure similar to the one of pyrrolidones, MI-77301 displays a different stereochemistry in the quaternary carbon atom and changed halogen substitution patterns in both phenyl rings, as well as a conformationally-constrained cyclohexanol group (Figure 4) [115,121]. This compound stabilizes p53 and activates its pathway, resulting in the interruption of the cell proliferation, cell cycle arrest and apoptosis, and is currently in phase I clinical trials for cancer treatment [122,123,124,125]. Another clinical trial was initiated with cancer patients involving safety and efficacy of MI-77301 [126] and Pimasertib (AS-703026), which is a highly selective, potent, ATP non-competitive allosteric inhibitor of MEK1/2 [127].

Back in 2011, Graves and co-workers reported the synthesis of a new spiroindoline MDM2 inhibitor, RO8994, which was designed combining the structural features of RG7388 and the spiroindolinone core structure of MI-888 [128,129]. Indeed, RO8994 showed to be as active as RG7388 and more than MI-888; the compound showed marked tumor growth inhibition of WTp53, MDM2-amplified SJSA-1 osteosarcoma tumor xenograft model, displaying significant tumor growth inhibition (>60%) at 1.56 mg kg^−1^, tumor stasis at 3.125 mg kg^−1^ and regression at 6.25 mg kg^−1^. The synthetic route towards RO8994 developed by Graves’ group initiates with a cycloaddition reaction between **80** and **81** promoted by silver fluoride, giving compound **82a** as intermediate (Scheme 21a). Next, **82a** is isomerized to **82b** in the presence of DBU via intermediate **83**. Finally, the deprotection of the Boc group leads to **84**, which after an amidation step, gives RO8994 as a racemic mixture. The desired enantiomer can be obtained via separation using chiral supercritical fluid chromatography (SFC). Some of the drawbacks of this synthetic pathway is the low overall yield (around 2%), which is a consequence of the formation of complex mixtures during the different steps and the need for SFC isolation of the enantiomers formed in the final step. Considering these factors, L. Shu and co-workers reported a new approach for the preparation of RO8994 (Scheme 21b). In this new approach, the cycloaddition reaction of **80** with **86** promoted by DBU gives **87** with high selectivity. Next, the hydrolysis of **87** affords acid intermediate *rac*-**88**, which undergoes a resolution step with *N,N*-dimethyl-((*R*)-1-phenylethyl)amine to give the enantiopure acid **88-***salt*. The desired product RO8994 is then formed after the amidation of **88** with ammonium hydroxide. This four-step procedure was able to generate enantiopure RO8994 with a 33% overall yield, and could be scaled up to produce up to multihundred grams of the product [130].

The strategy reported by L. Shu and co-workers was later used by Graves’ group to produce a new family of spiroindoline-type compounds containing bioisosteric replacements of the 6-chlorooxindole group in RO8994. This investigation led to the discovery of RO5353 and RO2468, which showed great potential for clinical development as highly potent, selective, and orally active p53-MDM2 inhibitors (Figure 6) [131].

Later in 2015, Ivanenkov’s group reported the regioselective synthesis of a series of compounds containing the spiro-indolinone scaffold from the one-pot reaction of commercially available 2-thioxoimidazol-4-one **89**, sarcosine **90** and isatin **91** (Scheme 22) [132]. Such molecules were evaluated on their ability induce apoptosis and block the proliferation of HepG2, Hek293, MCF-7, SiHa and HCT116 cell lines. Interestingly, the best activity was shown by compound **92a** which showed a comparable performance to Nutlin-3 against MCF-7 cell lines, with an IC_50_ value of 4.88 ± 1.5 µM.

Considering the high activity of spiropyrazoline oxindoles in the reactivation of WTp53, Santos’ group has recently described the synthesis of these scaffolds with a few structural modifications, such as the use of different substituents in the pyrazoline ring and their evaluation in the isogenic HCT116 cell lines pair, either in the presence or absence of TP53. The family of compounds **95** was synthetized through the 1,3-dipolar cycloaddition reaction between 3-methylene indoline-2-ones **93** and nitrile imines that were formed in situ via the dehydrohalogenation of hydrazonoyl chlorides **94**. Among the 18 compounds tested, one showed promising results (R^1^ = F, R^2^ = Ph, R^3^ = *t*Bu and R^4^ = Ph), against the cancer cell line expressing WT p53, with good activity and low cytotoxicity (Scheme 23) [133].

Very recently, two phase I clinical trials were initiated for the compound known as DS-3032b (Figure 7) aiming to study its safety, tolerability and pharmacokinetics in patients with hematological malignancies [134] and with advanced solid tumors or lymphomas [135]. DS-3032b appears to be a promising compound to treat hematological malignancies with MDM2 expression/amplification in leukemic blasts.

Another interesting MDM2 inhibitor is MK-8242. Although its structure is not available to date, it has been evaluated singly [136] in clinical trials and in combination with cytarabine in adult participants with refractory or recurrent acute myelogenous leukemia (AML) [136]. This compound is also well tolerated by patients with liposarcoma and other advanced solid tumors by inhibiting the binding of the MDM2 protein to the transcriptional activation domain of the WTp53 [137,138,139,140,141].

Other types of molecules containing the indole/indoline moiety have also shown promising activity in inhibiting the MDM2-p53 interaction. Ji and Li’s group, for instance, reported the synthesis of a family of (*E*)-3-benzylideneindolin-2-one derivatives **99** from the alkylation of 2,4-dihydroxibenzaldehyde **96** with alkyl iodines, giving **97**, followed by a Knoevenagel condensation with substituted indolinones **98** (Scheme 24) [142]. Among all of the synthetized compounds, **99a** presented the highest binding affinity to MDM2, as well as the highest antiproliferative activity against HCT116 (WTp53). Another interesting feature of this compound was its Ki value (0.093 mM), which can be translated into a high ligand efficiency when the amount of non-hydrogen atoms in the molecule is considered [40,41].

In 2006, Hardcastle, Lunec and co-workers reported the discovery of a series of indoline-type compounds as potent p53-MDM2 inhibitors by applying in silico screening and small library synthesis [143]. The compounds **105a** and **105b** showed promising biological activity results in SJSA cells, generating a dose-dependent increase of MDM2 and p21, which is related to the release of p53 transcriptional activity. The synthesis of the compounds was achieved starting from 2-benzoylbenzoic acid **100**, which underwent intramolecular cyclization, giving **101** (Scheme 25). Next, an amidation step leads to indoline **102**, which is then converted to indoline chloride **103**. After the reaction of **103** with benzyl mercaptan, **104** is formed and converted to the desired indoline product **105** after reacting with the appropriate alcohol. Later in 2011, the same group reported that slight structural changes led to even more active indolinone scaffolds [144].

Other related compound is JNJ-26854165, also known as serdemetan, a tryptamine derivative which has showed pronounced p53 reactivating properties. In 2011, Deutsch and co-workers reported the action of JNJ-26854165 against several types of human cancer cell lines; although a mechanism of action was not proposed, the authors verified that this compound also has potential as a photosensitizer [145]. Also in 2010, Andreef and co-workers reported that JNJ-26854165 was able to induce p53-mediated apoptosis in acute leukemia cells, as well as S-phase delay and E2F1-mediated apoptosis in p53 mutant cells, acting along with 1-β-arabinofuranosyl-cytosine or doxorubicin in order to cause p53-mediated apoptosis [146]. Later in 2011, Zhuang and co-workers described a series of pharmacodynamic and pharmacokinetic analyses that were conducted in order to identify the minimal effective dose of JNJ-26854165 in patients with advanced stage/refractory solid tumors [147]. A synthetic route towards JNJ-26854165 was described back in 2006 by Lacrompe and co-workers [148]; pyridine-functionalized amide **106** and fluoronitrobenzene **107** were used as starting materials to give coupling product **108** in the presence of a copper catalyst (Scheme 26). Next, a deprotection step in basic medium leads to **109**, which undergoes a Raney nickel-promoted reduction to furnish compound **110**. Finally, the reduction of indole **111** followed by its reaction with compound **110** in the presence of polymer-supported cyanoborohydride (PSCBH) in acidic medium leads to product JNJ-26854165.

In 2014, Buolamwini and Zhang’s group reported the activity of pyrido[*b*]indole SP-141 as a possible molecular-targeted chemotherapeutic agent [149]. SP-141 was able directly bind to MDM2, inhibit its expression and induce its autoubiquitination and proteasomal degradation. Importantly, this molecule has showed strong in vitro and in vivo antibreast cancer activity without evident host toxicity. The synthesis of SP-141 and several analogues was recently reported by Hiller and Mulcahy’s group, which evaluated such molecules as probes for the 5-hydroxytryptamine receptor (Scheme 27) [150]. The synthetic route starts with the Bischler-Napieralski reaction between indole **112** with triphosgene, giving intermediate **113**. **113** is then oxidized with DDQ, leading to pyridone **114**, which is next converted to the corresponding triflate **115**. Finally, the palladium-catalyzed Suzuki cross-coupling between **115** and boronic acid **116** gives rise to SP-141.

### 3.3. Pyrrol/Pyrrolidine, Piperidine, and Morpholine Derivatives

Based on MI-219 and its analog MI-77301, Graves and co-workers reported in 2013 the discovery of a second-generation MDM2 inhibitor with superior potency and selectivity, called as RO5503781 (also known as RG7388, Scheme 28) [151,152].

The key step in the synthesis of this compound is the formation of the pyrrolidine core through a 1,3-dipolar cycloaddition reaction between (*Z*)-α-cyanostilbene **119** (formed through the Knoevenagel condensation between **117** and **118**) and the imine **122** (obtained from the reaction between amine **119** and aldehyde **120**) promoted by AgF, giving substituted pyrrolidine intermediate **123**. After a Boc-deprotection step using standard conditions, affording **124**, and a subsequent amidation using HATU as coupling agent, the products were obtained as a racemic mixture. Next, a chiral SFC was used to separate the two enantiomers (Scheme 28). RG7388 has advanced to phase I clinical trials as a single agent or in combination with cytarabine in patients with acute myelogenous leukemia, neoplasms, and hematologic neoplasms [153].

Another class of compounds that has been successfully used as MDM2 inhibitors are piperidinones. Poppe and co-workers, for instance, reported in 2012 that a new class of piperidinones that are able to bind to the same three pockets as previously reported for the Nutlins, spiro-oxindoles and pyrrolidines, and also interacts in the *N*-terminal region of human MDM2, in particular Val14 and Thr16 [154]. Indeed, the synthesis and initial SAR of rigid cores capable of holding two aromatic rings in proximity had led Sun and co-workers closely before in the same year to the identification of a piperidinone derivative as a novel inhibitor of the MDM2-p53 protein–protein interaction [155]. Computational docking showed that the *p*-chloro substituted phenyl ring occupied the Trp23 pocket, while the *m*-chloro substituted one was in the Leu26 pocket and the cyclopropyl group occupied the Phe19 pocket. The carboxylic acid group at the 3-position of piperidinone also showed to have an additional interaction with the His96 residue of MDM2. In this sense, a series of optimizations led to a piperidinone derivative featuring a chiral *tert*-butyl 2-butanoate **126** which is 8-times more potent than **127** to MDM2 (Scheme 29a). The crystal structure of **126** showed that the two phenyl groups had a gauche-like orientation, which was less favored over the more stable anti-like conformation, as suggested by quantum mechanical calculations. Next, an extra methyl group was inserted at C3 along with further modification of the *tert*-butyl ester into a hydroxyl group, yielding AM-8553, which showed to be a highly potent, selective, and orally bioavailable inhibitor of MDM2-p53. The importance of the carboxylic acid moiety was confirmed by means of its removal from the molecule, which caused a 44-fold loss in binding affinity [156]. The same group also reported later the piperidone prototype AMG 232, which was created based on the structure of AM-8553 (Scheme 29a) [157].

The synthesis of AMG 232 was accomplished by initially reacting compound **127** with methyl methacrylate in the presence of a strong base, giving **128**. The asymmetric reduction of **128** afforded **129**, which next underwent an intramolecular cyclization step, furnishing lactone **130** (Scheme 29b). Lactone **130** was then alkylated with ally bromide to produce **131**, following an amidation reaction with valinol, giving **132**. After another intramolecular cyclization which leads to **133** and a thiol insertion/oxidation reaction, AMG 232 is obtained. AMG 232 is currently under clinical trials, in a phase 1b/2a study, for the evaluation of its activity in patients with metastatic melanoma in combination with trametinib and dabrafenib vs. trametinib combined with dabrafenib [158].

Rew and co-workers also reported later in 2014 the discovery of AM-7209, another piperidone that inhibits the MDM2-p53 interaction by inhibiting MDM2 [159]. This compound is structurally related to its predecessors, but the isopropylsulfonyl group was replaced by a butylsulfonyl group, and the isopropyl group for a cyclopropyl group in order to improve the interactions with the Phe19(p53) pocket in MDM2. Additionally, a *m*-fluoro group was inserted into the C6 phenyl ring, and the carboxylic acid was converted to a secondary amide. The structural changes led to a higher activity against the HCT-116 colorectal carcinoma xenograft model (SJSA-1 EdU IC50 = 1.6 nM). The synthesis of AM-7209 was accomplished using a fluorinated analogous of compound **131** (see the synthesis of **131** in Scheme 29) as a starting material (Scheme 30) [159]. After the conversion of **131** into amide **137** through the reaction with aminoalcohol **136**, **137** undergoes an intramolecular cyclization step in the presence of *p*-toluenesulfonic anhydride, leading to lactam **138**. **138** then furnishes bicyclic intermediate **139** after the reaction with *p*-toluenesulfonic acid, which next produces intermediate **140** in two steps. The carboxylic acid moiety in **140** is subsequently converted into an amide group via the reaction with 4-aminobenzoic acid **141** in the presence of EDC as a coupling agent, finally giving product AM-7209.

Another compound that has reached the clinical trials stage was CGM097, which was developed by Holzer and co-workers from Novartis and is being tested in patients with advanced solid tumors who have progressed despite standard therapy or for whom no standard therapy exists [160]. A Phase I dose escalation study of CGM097 in adult patients was also programed for selected patients with advanced solid tumors characterized by WTp53-dependent status [161,162]. The synthesis of CGM097 conducted using the convergent route is presented in Scheme 31.

The synthetic route starts with compound **142**, which initially undergoes an alkylation with iodopropane, giving **143**. Next, a carbonylation reaction leads to **144**, with a subsequent sulfonylation leading to **145**. An asymmetric reduction of the imine using an organotin compound and a rhodium complex as catalyst then gives **146**, which undergoes an intramolecular asymmetric cyclization step, furnishing fragment **147**. The synthesis of the other moiety of the molecule starts with a reductive amination of aldehyde **148**, giving **149**. An alkylation of the amine group gives **150**, which is submitted to a Boc deprotection step to furnish **151**, which leads to **152** upon a nucleophilic substitution step with methyl 2-bromoacetate. Subsequently, the reductive amination of aldehyde **153** with compound **152** leads to the formation of **154**, which is then submitted to an intramolecular cyclization reaction to produce **155**. Finally, a coupling reaction of **155** with fragment **147** affords the desired compound CGM097.

Besides piperidones, structurally-related morpholine derivatives have also revealed to be effective and selective MDM2-p53 inhibitors [163,164]. In 2014, Gonzalez and co-workers described the discovery of AM-8735, a potent and selective MDM2 inhibitor (SJSA-1 EdU IC50 = 25 nM) that showed remarkable pharmacokinetic properties and in vivo efficacy, also displaying considerable improvements in hepatocyte stability when compared to the related piperidinone inhibitors. The synthetic route towards AM-8735 starts with the conversion of 4-chlorobenzyl bromide **156** to nitroalkane **157**, which undergoes a Henry reaction with 3-chlorobenzaldehyde to furnish an amino alcohol, which subsequently undergoes an in situ protection to its corresponding silyl ether **158** (Scheme 32) [163]. Subsequently, the reduction of the nitro group in **158** leads to **159**, which after a chiral resolution step delivers the desired amino alcohol **160** with a 97% enantiomeric excess (*ee*). Alternatively, **160** can be produced from commercially available amino acids in five steps (not shown in Scheme 32). Next, morpholine **161** is obtained through the reaction of aminoalcohol **160** with chloroacetyl chloride in basic medium. **161** then undergoes an alkylation reaction with bromide **162** to furnish **163** as a mixture of interconvertible diastereoisomers. After separation, **163b** is reduced to alcohol **164** with superhydride and further converted to a thioether under Mitsunobu conditions. Finally, AM-8735 is obtained after an alkylation/oxidation sequence.

### 3.4. Pyrimidine and Quinazoline Derivatives

In recent years, pyrimidines and quinazoline-type molecules have also been pointed as promising p53-MDM2 inhibitors. Back in 2005, Weisman and co-workers have identified a series of 7-nitro-5-deaza-flavin compounds, the so-called HDM2 ligase inhibitor 98 (HLI98) [165]. Interestingly, the HLI98 compounds were somewhat specific for HDM2/MDM2, but effects on HECT E3s (ubiquitin ligase) both in vitro and in cells, were also observed. Fischer and co-workers have reported the synthesis of the HLI98 compounds starting from 2,4,6-trichloropyrimidine **165**, which was converted to 2-chlorouracil **166** via basic hydrolysis (Scheme 33) [166]. Next, the reaction of **166** with substituted anilines gives rise to 6-anilinouracils **167**, which upon condensation with 2-halo- or 2-tosylaldehydes **168**, furnish the products HLI98.

In 2009, high-throughput screening studies conducted by Allen and co-workers identified chromenotriazolopyrimidines as possible p53-MDM2 inhibitors [167]. In this case, rigid core of the compounds was believed to project the substituents into key p53 binding pockets of MDM2. Indeed, cellular assays proved the such compounds were able to to upregulate p53 protein and signaling, causing p53-dependent inhibition of proliferation and apoptosis. The synthesis of these compounds was achieved starting from substituted benzaldehyde **169** and 2′-hydroxyacetophenone **170**, which reacted in basic medium giving chalcone derivative **171** (Scheme 34). **171** then undergoes a cyclization with 3-amino-1,2,4-triazole **172**, leading to intermediate **173**, which undergoes cyclization with benzaldehyde derivatives to furnish **174**. Finally, the alkylation of **174** leads to **175** as a mixture of diastereoisomers.

HDM201 is another small molecule inhibitor of p53-MDM2 and is able to restore the normal function of WTp53, being inspired in the Nutlins family (Figure 8)[168]. This compound is under investigation in phase I clinical trials as an anticancer drug in patients with liposarcoma [169].

In 2011, Lu, Lim and co-workers reported their findings on the development of pyrrolopyrimidines which act as *R*-helix mimetics able to interrupt *R*-helix-mediated protein–protein interactions, acting as dual MDMX/MDM2 inhibitors [170]. Importantly, the group developed a solid-phase method for the divergent synthesis of a wide library of pyrrolopyrimidines (Scheme 35). The synthetic route towards pyrrolopyrimidines **184** starts with the bromoacetylation of Rink amide MBHA resin **176**, leading to **177**. Then, the bromine group was substituted by amines, giving **178**, which underwent the same procedures to give **179**. Next, **179** is reacted with 4,6-dichloro-2-(methylthio)-pyrimidine-5-carbaldehyde **180**, giving **181**, which undergoes intramolecular cyclization in the presence of DBU, giving **182**. The thioether group in **182** is then oxidized to the corresponding sulfone **183**, which underwent substitution with several amines followed by the cleavage of the resin-bound compounds with trifluoroacetic acid (TFA), giving pyrrolopyrimidines **184** as products.

### 3.5. Pyrazole and Pyrrolone Derivatives

MDMX is another key regulator of the p53 pathways, and shares homology with MDM2 in its p53-binding domains [171,172]. However, MDMX is thought to regulate p53 via different mechanisms: while MDM2 mostly regulates the stability and subcellular localization of p53, MDMX directly regulates p53 transcription. Like MDM2, MDMX is genetically amplified in several types of cancers, and therefore, the inactivation of both MDM2 and MDMX is believed to be crucial to induce an effective p53 response in tumor cells that express wild-type p53. In this sense, in 2010 Dyer and co-workers reported the identification and characterization of the first small molecule inhibitor of MDMX, the pyrazole named SJ-172550 [173]. This compound was able to bind reversibly to the p53-binding pocket of MDMX, displacing p53 and effectively killing retinoblastoma cells in which the expression of MDMX was enlarged. Importantly, when combined with the MDM2 inhibitor nutlin-3a, SJ-172550 showed superior cytotoxicity in cells that expressed wild-type p53. The synthesis of SJ-172550 was outlined by Russel, Huber and co-workers (Scheme 36) [174]. Initially, 3-ethoxy-4-hydroxybenzaldehyde **185** is chlorinated to form **186**, which is next submitted to a reaction with methyl 2-bromoacetate, giving methyl 2-(2-chloro-6-ethoxy-4-formylphenoxy)acetate **187** [175]. In parallel, phenyl hydrazine **188** is reacted with ethyl acetoacetate, forming pyrazolone **189**. The base-catalyzed reaction of **189** with fragment **187** gives rise to the product SJ-172550.

In 2013, Sheng, Zhang and co-workers reported the synthesis of a set of pyrrolo[3,4-*c*]pyrazole derivatives **197** that were excellent simultaneous inhibitors of p53-MDM2 and NF-κB (five DNA-binding proteins that are often hyperactive in cancer and inflammatory processes) [176]. The developed molecules were able to efficiently inhibit tumor growth in the A549 xenograft model. Additionally, compound **197** displayed great oral bioavailability (72.9%). The synthesis of such derivatives starts with the generation of methyl pyruvates **192** from the Claisen condensation of diethyl oxalate **191** and acetophenone **190** (Scheme 37). Next, the reaction of **192** with 4-bromobenzaldehyde **193** and *N*-(3-aminopropyl)imidazole **194** leads to **195**, which is then converted to pyrrolopyrazole **196** after reaction with hydrazine in acidic medium. Finally, the reaction of **196** with different haloalkanes gives rise to the desired products **197**.

In a similar path, the same group described later a series of pyrrolones **199** which showed to be striking activity as MDM2/MDMX dual inhibitors, impeding the tumor growth in the A549 xenograft model (Scheme 38) [177]. The compounds could be synthetized in a similar fashion to that of pyrrolopyrazoles **197**, but here intermediate **198** underwent a Mitsunobu reaction with a variety of alcohols to furnish the corresponding ether-pyrrolidone derivatives **199**.

### 3.6. Benzodiazepinone Derivatives

Benzodiazepinediones have also been studied with regards to their ability of reactivating p53 pathways. Using combinatorial chemistry techniques, Grasberger and co-workers identified compound DP222669 as antagonist of the HDM2-p53 interaction. The authors were also able to generate the X-ray crystal structure of the antagonists bound to HDM2, which revealed their *R*-helix mimetic properties. The synthesis of these compounds was achieved by using an Ugi multicomponent reaction with aldehydes **200**, *N*-Boc protected carboxylic acids **201**, amines **202** and isocyanide **203** to give intermediate **204**. The intramolecular cyclization of **204** leads to the formation of benzodiazepinediones **205** (Scheme 39) [178].

Later in 2011, Sheng, Zhang and co-workers described the synthesis of the thio-benzodiazepines and their biological activity of p53-MDM2 inhibitors [179]. Most compounds had nano/micromolar affinity towards MDM2, and some showed activity similar to that of nutlin-3a. Compound **211a** (R = 4-CF_3_C_6_H_4_), in particular, exhibited a high antitumor activity against the U-2 OS human osteosarcoma cell line (IC50 = 1.06 mM). These molecules were synthetized through the Ugi multicomponent reaction of amine **206**, benzaldehyde **63**, isocyanide **208** and substituted 2-nitrobenzoic acids **207**, giving bisamide **209** as intermediate (Scheme 40). Next, benzodiazepinone **210** is obtained after the reduction of **209** with iron powder followed by intramolecular cyclization, and is converted to **211** after a reductive amination step. Finally, benzodiazepinones **210** are converted to the corresponding thio-benzodiazepinones **211** after reacting with the Lawesson’s reagent.

### 3.7. Benzofuroxan and Furan Derivatives

In 2010, Yan’s group described the use of benzofuroxan derivative XI-006 (NSC207895) as an inhibitor of the MDMX expression in cancer cells; the treatment of MCF-7 cells with this compound activated p53, resulting in a higher expression of proapoptotic genes [180]. Interestingly, the compound also worked additively with nutlin-3a to activate p53. Later in 2015, Pishas and co-workers reported that XI-006 was able to induce p53-independent apoptosis in Ewing’s sarcoma [181]. The synthesis of XI-006 can be achieved starting with the base-catalyzed intramolecular cyclization of nitroaniline **213**, giving **214** (Scheme 41) [182]. **214** then undergoes a nitration step, giving **215**, which next is isomerized to **216** in acidic medium. The reaction of **216** with piperazine leads to **217**, and a methylation step delivers the product XI-006.

Another valuable example is the compound called RITA (NSC 652287), which displays a very simple structure (Scheme 42). This compound induces both DNA-protein and DNA-DNA cross-links with no detectable DNA single-strand breaks, and has also been shown to inhibit the p53-MDM2 interaction as well as target p53 mutants R175H, R248W, R273H, and R280K, restoring transcriptional activity and inducing apoptosis [183,184]. The synthesis of RITA may be accomplished starting with the copper(I)-catalyzed homocoupling of terminal alkyne **218**, giving 1,3-butadiyne **219**, which next undergoes a cyclocondensation with water in basic medium, giving rise to the product [185] (Scheme 42).

## 4. Small Molecules That Reactivate the p53 Protein through Other Pathways

Many compounds reported in the literature have shown the ability to restore the activity of p53 but apparently do not act through the aforementioned pathways. Leptomycin B (LMB), for instance, is a *Streptomyces sp.* metabolite that induces the transcriptional activity of a p53-dependent reporter plasmid [186]. Along with this LMB activity, an p53-dependent increase in the levels of the products of two p53-dependent genes, p21 and MDM2 were also observed [187]. Overall, LMB is believed to function mostly as an inhibitor of the export of proteins from the nucleus to the cytoplasm owing to its ability to interact with, and impair the function of the nuclear export factor CRM-1 [188] In this sense, LMB is a very effective inducer of the p53 response which does not act directly through the DNA damage pathway [189]. Indeed, LMB has showed to be effective against human lung adenocarcinoma cell lines, among others [190]. Other LMB derivatives have also showed to be efficient nuclear export inhibitors, having showed promising efficacy in multiple mouse xenograft models [191]. The total synthesis of leptomycin B was reported back in 1998 by Kobayashi and co-workers (Scheme 43) [192]. Initially, *E*-crotyl alcohol **220** undergoes a Sharpless oxidation reaction, giving epoxide **221**. Next, a regioselective ring-opening leads to **222**, which after a carboxylation/thermal treatment gives rise to lactone **223**. Next, the lactone **223** is protected as isopropyl acetal **224**, and after the deprotection of the benzyl group and Swern oxidation, aldehyde **225** is formed. **225** then undergoes a Wittig reaction with fragment **226**, affording conjugated diene **227**. Then, **227** is returned to its lactone form **228**. In parallel, carboxylic acid **229** (formed from the ozonolysis of geraniol, not shown in Scheme 43) is condensed with a chiral oxazolidinone and submitted to a methylation step to give **230**, which after removal of the chiral auxiliary and reduction, leads to aldehyde **231**. Next, **231** undergoes an Evans aldol reaction, giving **232**, which undergoes removal of the auxiliary group protection of the OH group, giving **233**. A second aldol condensation leads to **234**, which once again undergoes the removal of the chiral auxiliary and a reduction step, leading to **235**. After the exchange of the terminal protecting group from MOM to TBS, fragment **236** is formed. Finally, the fragments **228** and **236** are combined, giving **237**, which next undergoes removal of the alcohol protection group leading to **238**, and an oxidation step gives leptomycin B.

Another efficient approach for the activation of p53 is the inhibition of casein kinase 1A1 (CKIa). Indeed, this method was successfully employed by several groups, with promising results in vivo in mouse models of intestinal cancer [26] and in cultured leukemia cells [193]. In 2018, Neriah’s group reported their findings on the use of a series of molecules, among them A51, in primary mouse acute myeloid leukemia cells [194]. Interestingly, the blocking of CKIa along with CDK7 and/or CDK9 synergistically stabilizes p53, depriving leukemia cells of survival and proliferation, maintaining SE-driven oncogenes and inducing apoptosis. The synthesis of A51 starts with 1-methylpyrazole **239**, which undergoes deprotonation by a strong base and adds to cyclopropanecarboxaldehyde, giving **240** (Scheme 44). Next, **240** is dehydroxylated to produce **241**, which next undergoes a bromination step giving **242**. In sequence, **242** undergoes treatment with a strong base followed by the reaction with 2-isopropoxy-4,4,5,5-tetramethyl-1,3,2-dioxaborolane, giving **243**, which is then is reacted with 2,4,5-trichloropyrimidine, giving substitution product **244**. Finally, the reaction of with **244** with *trans*-diaminocyclohexane under microwave irradiation delivers the product A51.

The human genome encompasses seven sirtuins (SirT1-7), and those are implicated in a series of essential cellular processes and disease conditions [195]. SirT1, especially, is involved in the regulation of the transcriptional function of p53, while SirT2 is involved in the direct deacetylation of α-tubulin and histone H4 [196,197,198,199]. Considering these facts, Lain and co-workers identified back in 2008 a small-molecule activator of p53 at low micromolar concentrations [200]. The authors were able to define the mechanism of action for this compound, which reversibly increased p53 and p21CIP/WAF1 protein levels and decreased cellular growth in several types of tumor cell lines. Indeed, tenovin-1 has shown similar potency in both mammalian cell-based assays and melanoma xenograft tumors. On the other hand, the low solubility of tenovin-1 in water led to the development of a molecule with higher water solubility, tenovin-6. Remarkably, tenovin-6 has also displayed a slightly higher potency in p53 activation [200]. The synthesis of such compounds can be accomplished by the initial conversion of *p*-tert-butylbenzoic acid **245** to the corresponding acyl chloride **246** (Scheme 45) [201]. Next, **246** is reacted with sodium thiocyanate and subsequently with phenylenediamine, giving thiourea **247**. Finally, the reaction of **247** with the appropriate acyl chloride delivers the product.

Another molecule that has shown to be a promising SirT1 inhibitor is inauhzin, as reported by Lu’s group [202]. This triazino[5,6-*b*]indol-phenothiazine-type molecule reactivates p53 by inhibiting SIRT1 activity and promoting p53-dependent apoptosis of cancer cells without provoking genotoxic stress. Structural changes in inauhzin have also led to analogues with promising p53 reactivating properties [203]. For the synthesis of inauhzin, 5*H*-[1,2,4]triazino[5,6-*b*]indole-3-thiol **250** is initially synthetized from commercially available isatin **248** and thiosemicarbazide **249** (Scheme 46) [203]. In parallel, bromide **254** is obtained from the reaction between **251** and 2-bromobutyryl bromide **252**. Finally, fragments **250** and **254** undergo a reaction in basic medium to furnish inauhzin.

Recently, El-Deiry and co-workers found that the imidazole-type molecule CB002 is a p53 pathway-restoring compound that induces tumor selective cell death via the induction of NOXA, a pro-apoptotic protein, being able of targeting mutant versions of p53 for MDM2-independent proteasomal degradation. Indeed, this compound was able to induce several p53 target genes in various mutant p53 cell lines without affecting their p53 protein turnover. The synthesis of CB002 was reported in 1994 by Kokel, which used 6-amino-1,3-dimethyluracil **255** as a starting material in the presence of (dichloromethylene)dimethylammonium chloride (Scheme 47) [204]. Next, the reaction of **256** with azidotrimethylsilane leads to **257**, which then undergoes an addition-elimination step with aniline to give **258**. Finally, **258** undergoes a base-catalyzed intramolecular step to furnish 8-anilinotheophylline CB002 as product.

## 5. Conclusions and Outlook

In the last decades, a lot of groundbreaking work was conducted regarding the development of small molecules for p53 reactivation, and two main pathways have been especially highlighted: (1) Molecules that can react covalently with the thiol groups in the cysteine residues of mutant p53, which leads to conformational changes that can reactivate the protein and (2) molecules that are able to disrupt the MDM2-p53 protein complex, releasing the WTp53 protein to resume its function. In this sense, a lot of advanced studies were published and demonstrated that comprehending the structure of p53 is crucial for achieving its reactivation.

A recent breakthrough was the discovery that mutant p53 undergoes aggregation in a very similar way to that observed with other amyloid proteins [11,25,27], playing a crucial role in the development of cancer through loss of p53 function, a dominance-negative effect and gain of oncogenic function. Compounds and peptides that have been described to inhibit mutant p53 aggregation also lead to a decline on tumor proliferation and migration. Thus, the misfolded and aggregated states of mutant p53 have become highly promising targets for the development of novel therapeutic strategies against cancer [11].

On the other hand, although much has already been accomplished, there is still much to achieve, especially on the development of synthetic methods to reach the promising molecules. In addition, the use of computational approaches along with in vitro experiments will aid in the design of new drugs [205,206,207,208]. For instance, Lukman and coworkers have demonstrated by simulations how loop 1 of p53 DNA binding domain is the most dynamic segment among the DNA-contacting loops and how it can be used as a potential target [205].

The molecules tested so far have not yet reached the market and the effects can still be improved in terms of lower toxicity to normal cells, potency and affinity to targets. Also, many of the papers discussed in this review still employ outdated stoichiometric methods, toxic solvents and harsh reaction conditions, and such approaches can be easily substituted by greener ones. However, the growing awareness regarding environmental issues in the last few years have generated a change in the design of synthetic methods, which lead us to believe that this pressing change will come in due time.

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
