# Peer review of "Recent Synthetic Approaches towards Small Molecule Reactivators of p53"

_biomolecules, 2020, doi:10.3390/biom10040635_

Round 1

Reviewer 1 Report

The work is detailed and of interest for experts of the field. Needs improved language fluidity in the Introduction sections of the manuscript, especially when switching from one topic to the next.

Mainly the point of this review is focused on the detailed chemical preparations and strategies  used to obtain the molecules discussed. It is relevant and interesting, to my opinion, especially for the experts of the field relative also to future perspectives as improvement of procedures.

It is original in the way that it chemically discuss arguments already exstensively discussed in literature, it gives a different point of view.

For a non expert of the field it is not easy to read. The background must be discussed in a more complete way as to insert missing strategies to activate p53 function but at the same time must be kept simple and fluid.

Author Response

Reviewer 1.

  1. The work is detailed and of interest for experts of the field. Needs improved language fluidity in the Introduction sections of the manuscript, especially when switching from one topic to the next.

Mainly the point of this review is focused on the detailed chemical preparations and strategies  used to obtain the molecules discussed. It is relevant and interesting, to my opinion, especially for the experts of the field relative also to future perspectives as improvement of procedures

R.: Thank you. We have made changes in the Introduction section as recommended by the reviewer.

  1. For a non expert of the field it is not easy to read. The background must be discussed in a more complete way as to insert missing strategies to activate p53 function but at the same time must be kept simple and fluid.

R.: We have carefully revised the manuscript to make it easier to read. We reinforced the background. We changed the subtitles and grouped the molecules according to the structures within each type of inhibitor.

Reviewer 2 Report

The review, according to the abstract, is focused on currently adopted approaches for activation and reactivation of the p53 tumor suppressor function, and on the synthetic approaches that are involved in the development and preparation of such small molecules. If that is the case, the title is not very accurate as it only mentions reactivators of mutant p53. If the review is focused on mutant p53 reactivators, the section about MDM2 inhibitors that activate wild-type p53, not mutated p53, should be removed.

MAJOR CORRECTIONS:

-In this manuscript several important chemical families known to target mutated p53 are missing, as the authors only focused on molecules that can react covalently with the thiol groups in the cysteine residues of mutant p53.  Those works should be included in the review.

-Several very good reviews recently published on small molecules that activate the p53 tumor suppressor function (focused on MDM2 inhibitors, dual MDM2/X inhibitors or mutant p53 reactivators) need to be cited in the introduction if this review is accepted for publication.

-Comparing to other reviews in the same field, the main innovation of this manuscript is the discussion on the synthetic approaches involved in the preparation of the molecules. So, with a different organization, this review would be more suitable for an organic chemistry journal.

Author Response

Reviewer 2.

  1. The review, according to the abstract, is focused on currently adopted approaches for activation and reactivation of the p53 tumor suppressor function, and on the synthetic approaches that are involved in the development and preparation of such small molecules. If that is the case, the title is not very accurate as it only mentions reactivators of mutant p53

R.: Thank you for the suggestion. Accordingly, we changed the title because the review is not only about reactivating mutant p53, but also wild-type (MDM2 inhibitors).

  1. In this manuscript several important chemical families known to target mutated p53 are missing, as the authors only focused on molecules that can react covalently with the thiol groups in the cysteine residues of mutant p53.  Those works should be included in the review.

R.: We have followed the suggestions of Reviewer 2 by including several chemical families to target mutant p53. Several references (with the respective schemes) have been added, both in the mutant and wild-type part, as well as other types of inhibitors. We have also added the seminal zinc paper "suggested" by the editor.

  1. Several very good reviews recently published on small molecules that activate the p53 tumor suppressor function (focused on MDM2 inhibitors, dual MDM2/X inhibitors or mutant p53 reactivators) need to be cited in the introduction if this review is accepted for publication.

R.: Several reviews were added as a reference, as suggested by Reviewer 2, mainly in the MDM2 part. As mentioned above, several new references were included.

  1. Comparing to other reviews in the same field, the main innovation of this manuscript is the discussion on the synthetic approaches involved in the preparation of the molecules. So, with a different organization, this review would be more suitable for an organic chemistry journal.

R.: We improved the organization of the manuscript as recommended. We have made the description of the synthetic approaches clearer and changed the order of appearance of the different classes of inhibitors.

We would again like to thank you, the Editor, and the Reviewers for your helpful suggestions and comments, which have greatly helped us improve our manuscript.

Round 2

Reviewer 2 Report

The authors have submitted a new version with a more adequate title. However, several reviews also based on small molecule p53 activators continue not be cited (e.g. Current Topics in Medicinal Chemistry, 2018, 18, 647-660; recent patents on anti-cancer drug discovery, 2019, 14, 324-369; Current Medicinal Chemistry, 2019, 26, 7323-7336, etc), and some of them were published by MDPI (e.g. Pharmaceuticals 2016, 9, 2, 25; Biomolecules 2020, 10, 303; etc). Moreover, some relevant scaffolds targeting DNA-binding mutated p53 were not included in the review. There is a mixture between compounds that act directly on wt p53, mut p53, MDM2 and MDMX and compounds that activate the p53 pathway indirectly. The review should be better organized before being accepted for publication.

Author Response

We are thankful to all of the reviewers for their careful and critical reading of our initial submission. We have revised the manuscript according to the editorial requests and comments of the reviewers. Enclosed is the revised version of our manuscript entitled “Recent synthetic approaches towards small molecule reactivators of p53” complying with the Biomolecules guidelines and a point-by-point reply to all raised comments (all the changes are highlighted in the text).

Reviewer 2.

“The authors have submitted a new version with a more adequate title. However, several reviews also based on small molecule p53 activators continue not be cited (e.g. Current Topics in Medicinal Chemistry, 2018, 18, 647-660; recent patents on anti-cancer drug discovery, 2019, 14, 324-369; Current Medicinal Chemistry, 2019, 26, 7323-7336, etc), and some of them were published by MDPI (e.g. Pharmaceuticals 2016, 9, 2, 25; Biomolecules 2020, 10, 303; etc). Moreover, some relevant scaffolds targeting DNA-binding mutated p53 were not included in the review. There is a mixture between compounds that act directly on wt p53, mut p53, MDM2 and MDMX and compounds that activate the p53 pathway indirectly. The review should be better organized before being accepted for publication.”

R.: Thank you. We have included all the references suggested by the reviewer with the appropriate text (e.g. Current Topics in Medicinal Chemistry, 2018, 18, 647-660; recent patents on anti-cancer drug discovery, 2019, 14, 324-369; Current Medicinal Chemistry, 2019, 26, 7323-7336, etc), and some of them were published by MDPI (e.g. Pharmaceuticals 2016, 9, 2, 25; Biomolecules 2020, 10, 303; etc).

            We have also included the suggestion to include the scaffolds targeting DNA-binding mutated p53. Accordingly, changes in the text were made (highlighted in the manuscript).

Again, we would like to thank the Editor and the Reviewers for the helpful suggestions and comments, which have greatly improved our manuscript.